# Consortium of *Lactobacillus crispatus* 2029 and *Ligilactobacillus salivarius* 7247 Strains Shows In Vitro Bactericidal Effect on *Campylobacter jejuni* and, in Combination with Prebiotic, Protects Against Intestinal Barrier Dysfunction

**DOI:** 10.3390/antibiotics13121143

**Published:** 2024-11-28

**Authors:** Vyacheslav M. Abramov, Igor V. Kosarev, Andrey V. Machulin, Evgenia I. Deryusheva, Tatiana V. Priputnevich, Alexander N. Panin, Irina O. Chikileva, Tatiana N. Abashina, Ashot M. Manoyan, Olga E. Ivanova, Tigran T. Papazyan, Ilia N. Nikonov, Nataliya E. Suzina, Vyacheslav G. Melnikov, Valentin S. Khlebnikov, Vadim K. Sakulin, Vladimir A. Samoilenko, Alexey B. Gordeev, Gennady T. Sukhikh, Vladimir N. Uversky, Andrey V. Karlyshev

**Affiliations:** 1Federal Service for Veterinary and Phytosanitary Surveillance (Rosselkhoznadzor) Federal State Budgetary Institution “The Russian State Center for Animal Feed and Drug Standardization and Quality” (FGBU VGNKI), 123022 Moscow, Russia; 2Kulakov National Medical Research Center for Obstetrics, Gynecology and Perinatology, Ministry of Health, 117997 Moscow, Russia; t_priputnevich@oparina4.ru (T.V.P.); a_gordeev@oparina4.ru (A.B.G.);; 3Skryabin Institute of Biochemistry and Physiology of Microorganisms, Federal Research Center “Pushchino Scientific Center for Biological Research of Russian Academy of Science”, Russian Academy of Science, 142290 Pushchino, Russia; 4Institute for Biological Instrumentation, Federal Research Center “Pushchino Scientific Center for Biological Research of Russian Academy of Science”, Russian Academy of Science, 142290 Pushchino, Russia; 5Blokhin National Research Center of Oncology, Ministry of Health, 115478 Moscow, Russia; irinatchikileva@mail.ru; 6Alltech Company, 105062 Moscow, Russia; 7Federal State Budgetary Educational Institution of Higher Education, St. Petersburg State University of Veterinary Medicine, 196084 Saint Petersburg, Russia; 8Gabrichevsky Research Institute for Epidemiology and Microbiology, 125212 Moscow, Russia; 9Institute of Immunological Engineering, 142380 Lyubuchany, Russia; 10Department of Molecular Medicine, Morsani College of Medicine, University of South Florida, Tampa, FL 33612, USA; vuversky@usf.edu; 11Department of Biomolecular Sciences, School of Life Sciences, Chemistry and Pharmacy, Faculty of Health, Science, Social Care and Education, Kingston University London, Kingston upon Thames KT1 2EE, UK; a.karlyshev@kingston.ac.uk

**Keywords:** *Campylobacter*, multidrug resistance, synbiotics, bactericidal activity, intestinal homeostasis

## Abstract

**Background/Objectives:** *Campylobacter jejuni* (CJ) is the etiological agent of the world’s most common intestinal infectious food-borne disease, ranging from mild symptoms to fatal outcomes. The development of innovative synbiotics that inhibit the adhesion and reproduction of multidrug-resistant (MDR) CJ in animals and humans, thereby preserving intestinal homeostasis, is relevant. We have created a synbiotic based on the consortium of *Lactobacillus crispatus* 2029 (LC2029), *Ligilactobacillus salivarius* 7247 (LS7247), and a mannan-rich prebiotic (Actigen^®^). The purpose of this work was to study the in vitro anti-adhesive and antagonistic activities of the created synbiotic against MDR CJ strains, along with its role in preventing intestinal barrier dysfunction, which disrupts intestinal homeostasis. **Methods:** A complex of microbiological, immunological, and molecular biological methods was used. The ability of the LC2029 and LS7247 consortium to promote intestinal homeostasis in vitro was assessed by the effectiveness of controlling CJ-induced TLR4 activation, secretion of pro-inflammatory cytokines, development of intestinal barrier dysfunction, and production of intestinal alkaline phosphatase (IAP). **Results:** All MDR CJ strains showed marked adhesion to human Caco-2, pig IPEC-J2, chicken CPCE, and bovine BPCE enterocytes. For the first time, we found that the prebiotic and cell-free culture supernatant (CFS) from the consortium of LC2029 and LS7247 strains exhibit an additive effect in inhibiting the adhesion of MDR strains of CJ to human and animal enterocytes. CFS from the LC2029 and LS7247 consortium increased the permeability of the outer and inner membranes of CJ cells, which led to extracellular leakage of ATP and provided access to the peptidoglycan of the pathogen for the peptidoglycan-degrading bacteriocins nisin and enterolysin A produced by LS7247. The LC2029 and LS7247 consortium showed a bactericidal effect on CJ strains. Co-cultivation of the consortium with CJ strains resulted in a decrease in the viability of the pathogen by 6 log. CFS from the LC2029 and LS7247 consortium prevented the growth of CJ-induced TLR4 mRNA expression in enterocytes. The LC2029 and LS7247 consortium inhibited a CJ-induced increase in IL-8 and TNF-α production in enterocytes, prevented CJ-induced intestinal barrier dysfunction, maintained the transepithelial electrical resistance of the enterocyte monolayers, and prevented an increase in intestinal paracellular permeability and zonulin secretion. CFS from the consortium stimulated IAP mRNA expression in enterocytes. The LC2029 and LS7247 consortium and the prebiotic Actigen represent a new synergistic synbiotic with anti-CJ properties that prevents intestinal barrier dysfunction and preserves intestinal homeostasis. **Conclusions:** These data highlight the potential of using a synergistic synbiotic as a preventive strategy for creating feed additives and functional nutrition products based on it to combat the prevalence of campylobacteriosis caused by MDR strains in animals and humans.

## 1. Introduction

*Campylobacter jejuni* (CJ) is the main cause of bacterial gastroenteritis of food origin worldwide, with various socio-economic consequences [1,2,3,4,5,6,7]. The disease occurs in all age groups, but children aged from birth to 5 years are more susceptible to the pathogen [8,9]. CJ colonizes the gastrointestinal tract (GIT) of food-producing animals such as poultry, pigs, and cattle [10,11,12] and is able to survive in soil, water, and unpasteurized milk [13,14,15,16]. CJ has the ability to survive in *Acanthamoeba*, which is important for its spreading in the environment and subsequent infection [17,18]. CJ refers to nosocomial infections that can develop into a viable but non-culturable (VBNC) form [8,19,20]. Numerous data indicate that poultry are the main source of human CJ infection [9,21,22,23,24].

Clinical manifestations of campylobacter infection in humans may include watery or bloody diarrhea, accompanied by abdominal cramps, nausea, fever, and vomiting [25,26,27]. Severe disorders, such as bacteremia, urinary tract infections, sepsis, and neuropathy (Guillain–Barré syndrome), can develop as complications of campylobacteriosis [28,29,30,31,32]. In newborns, CJ infection induces necrotizing enterocolitis (NEC), which has the highest infant mortality rate [33]. CJ is the cause of 96 million cases of intestinal infection worldwide each year [7,34,35]. In the European Union (EU), CJ is the cause of 9 million cases of disease, with an economic loss of about EUR 2.4 billion per year (https://www.efsa.europa.eu/en/topics/topic/campylobacter (accessed on 1 October 2024)). In the United States (US), CJ is the cause of 1.5 million human infections every year (https://www.cdc.gov/campylobacter/about/index.html (accessed on 1 October 2024)) with economic losses ranging from USD 1.3 to 6.8 billion per year. The World Health Organization (WHO) has listed campylobacter as a high-priority pathogen [36,37].

The decoding of the complete genome sequence of CJ isolated from the feces of a patient with diarrhea marked a new era in the study of the pathogenesis mechanisms of this common intestinal infection [38]. Currently, the use of genomic sequencing is the main approach for studying the epidemiology of CJ [39].

A polysaccharide capsule was discovered that provides the pathogen with the means for evasion from the immune control of the host organism, and the properties of this capsule were studied [40,41,42,43,44,45,46]. The capsular polysaccharide (CPS) of CJ forms an outer layer on the cell surface, which contributes to the virulence of the pathogen [47]. CJ has a set of fitness and virulence factors that help the bacterium escape the protective mechanisms created by the host [48,49,50,51,52,53,54,55].

Previously, campylobacteriosis in poultry was prevented by vaccination and treated with antibiotics and phages [56,57,58]. Macrolides and fluoroquinolones were the first- and second-choice alternatives, respectively, for the antimicrobial treatment of enteritis induced by CJ in humans.

The widespread use of antibiotics in animal husbandry has stimulated the emergence of antibiotic-resistant pathogens that are transmitted to humans along the food chain. Since the late 1980s, resistance to macrolides and fluoroquinolones has emerged [59].

This has led to the emergence of multidrug-resistant (MDR) strains and has created difficulties in the treatment of campylobacteriosis [60,61]. Numerous clinical studies have shown that patients infected with quinolone- or erythromycin-resistant strains of CJ are significantly more likely to experience complications compared with patients infected with a strain of the pathogen that is sensitive to quinolones and erythromycin [62,63,64].

MDR strains of CJ have appeared in many countries of the world [37,65,66,67,68,69,70,71,72]. Currently, the number of MDR strains of CJ is constantly increasing in both developed and developing countries, including Italy [39,73,74,75], China [76,77], the USA [78], Switzerland [79], South Korea [80,81], Serbia [82], Iran [83], Tunisia [84], South Africa [85,86,87], Kenya [88].

The growing threat to public health has led to the need to limit the use of antibiotics in animal husbandry and begin developing alternative strategies to combat this pathogen [89].

In the phage therapy of CJ infections, there is a tendency for the effectiveness of the drug to decrease during the course of treatment [90]. This indicates that CJ is able to quickly adapt to phage treatment due to the development of resistance.

At present, there is no approved vaccine for use in humans and farm animals, despite numerous studies on the development of vaccines against CJ [91,92,93,94,95,96]. Due to the ability of this pathogen to cause autoimmune diseases, the introduction of live attenuated vaccines for humans is undesirable [97]. Alternative strategies using probiotics, prebiotics, and synbiotics for the effective control of CJ colonization in the GIT of animals and humans are of increasing interest [98,99,100,101].

Despite the promising results [102,103,104,105,106], effective and reliable alternative approaches have yet to be developed. Currently, it is important to create synbiotics based on a consortium of probiotic strains that are tolerant to gastric and intestinal stresses, have high adhesive activity to animal and human enterocytes, and produce factors that inhibit adhesion and increase the permeability of the outer and inner membranes of CJ for penetration of peptidoglycan (PG)-degrading factors into the cell. Such synbiotics should include a prebiotic that enhances the anti-adhesive properties of the probiotic consortium. *Lactobacillus crispatus* strain 2029 (LC2029) was previously obtained from the human vagina [107], and *Ligilactobacillus salivarius* 7247 (LS7247) from the human intestine [108]. The genome of LC2029 contains gene clusters that are responsible for the production of helveticin-M, a class III bacteriocin, and the S-layer protein Slp2 as OM permeabilizers of intestinal pathogens. [109]. The genome of LS7247 contains gene clusters that are responsible for the production of the OM permeabilizer lactic acid for intestinal pathogens, the lantibiotic nisin, belonging to class I bacteriocins, and enterolysin A, belonging to bacteriocins of class IIIa [110]. Lectin-like proteins are expressed on the surface of CJ, recognizing mannose residues that play an important role in the adhesion and colonization of host enterocytes by the pathogen [111]. Significant reductions in the adherence of CJ strains to Hep-2 cells were observed in the presence of mannan oligosaccharides [112]. In vivo experiments on broiler chickens showed that the prebiotic Actigen^®^ (Alltech, Inc., Nicholasville, KY, USA), which is a yeast-derived mannan-rich fraction, inhibited CJ’s colonization of the cecum [113].

We have created a synbiotic based on the consortium of human vaginal LC2029 and human intestinal LS7247 strains and the prebiotic Actigen. The purpose of this work was to study the in vitro anti-adhesive and antagonistic activities of the created synbiotic against MDR CJ strains, along with its role in preventing intestinal barrier dysfunction that disrupts intestinal homeostasis.

## 2. Results

### 2.1. The Effect of the Prebiotic Actigen and CFS from the LC2029 and LS7247 Consortium on CJ’s Adhesion to Enterocytes

We found that all CJ strains showed marked adhesion to human Caco-2 (Table 1), pig IPEC-J2 (Table 2), chicken CPCE (Table 3), and bovine BPCE (Table 4) enterocytes. The adhesion index (AI) of the CJ strains was in the range of 21.6 ± 1.4–56.2 ± 1.7. Pretreatment of CJ strains with the prebiotic Actigen alone or the CFS of the LC2029 and LS7247 consortium alone reduced the AI to 5.8 ± 0.2–17.6 ± 0.9 (*p* < 0.01) and to 5.2 ± 0.7–11.5 ± 0.6 (*p* < 0.01), respectively. Pretreatment of CJ strains with a mixture of CFS + Actigen had an additive effect, reducing the AI to 1.1 ± 0.2–2.8 ± 0.6 (*p* < 0.001). The CFS from the consortium of LC2029 and LS7247 strains pretreated with proteinase K did not compete with CJ strains for adhesion to enterocytes. This indicates that the CFS component, which has the ability to effectively inhibit the adhesion of CJ strains to human and animal enterocytes, has a peptide or protein nature.

### 2.2. The Effect of Slp2 from LC2029 and the Prebiotic Actigen on CJ’s Adhesion to Caco-2 Enterocytes

The ability of Slp2 from LC2029—alone and together with the prebiotic Actigen—to inhibit CJ’s adhesion to Caco-2 enterocytes was studied. The results of these studies are shown in Table 5. Pretreatment of CJ strains with Slp2 alone or the prebiotic Actigen alone reduced the adhesion index of the CJ strains to Caco-2 cells (*p* < 0.01). Pretreatment of CJ strains with a mixture of Slp2 + Actigen had an additive effect on reducing the adhesion index of the CJ strains to Caco-2 cells (*p* < 0.001). Slp2 alone, after incubation together with proteinase K (∆Slp2), lost the ability to inhibit the adhesion of CJ strains to Caco-2 enterocytes, and it did not enhance the anti-adhesive effect of the prebiotic Actigen. We assume that the molecular factor providing the CFS with the ability to effectively inhibit the adhesion of CJ strains to human and animal enterocytes was Slp2.

### 2.3. The Effect of CFS from the LC2029 and LS7247 Consortium on ATP Leakage in CJ Cells

ATP leakage was investigated as non-selective pore formation and indices of CJ cell injury. CFS from the LC2029 and LS7247 consortium induced CJ cell damage and ATP leakage. The results of these studies are shown in Table 6. Cultivation of CJ strains with CFS increased the level of extracellular ATP from 5.2 ± 0.3–7.4 ± 0.9 (nm/OD) to 42.3 ± 1.1–85.4 ± 3.2 (nm/OD) (*p* < 0.01). The level of extracellular ATP induced by the CFS did not depend on the source of CJ strain isolation.

### 2.4. The Effects of the LC2029 and LS7247 Consortium on CJ Strains

The antibacterial activity of the LC2029 and LS7247 consortium against MDR CJ isolates is shown in Table 7. Cultivation of the LC2029 and LS7247 consortium for 48 h together with CJ isolates obtained from the feces of a pregnant woman, feces of a neonatal baby with NEC, blood and feces of a man, and feces of broilers, pigs, and cows reduced the levels of living MDR CJ cultures by 6 log. Thus, the LC2029 and LS7247 consortium had an antibacterial effect on MDR CJ pathogens regardless of their habitat.

### 2.5. The Effect of CFS from the LC2029 and LS7247 Consortium on TLR4 mRNA Expression in Caco-2 Enterocytes

CFS from the LC2029 and LS7247 consortium inhibited the increase in CJ-induced TLR4 mRNA expression in Caco-2 enterocytes (Table 8). All studied CJ strains significantly stimulated the expression of TLR4 mRNA in intestinal epithelial cells (*p* < 0.001). However, pre-cultivation of CJ strains with CFS and their subsequent introduction into the enterocyte culture medium canceled the increase in TLR4 mRNA expression.

### 2.6. The Effects of the LC2029 and LS7247 Consortium on IL-8 and TNF-α Production in Caco-2 Enterocytes

CJ strains stimulated IL-8 and TNF-α production in Caco-2 enterocytes (Table 9 and Table 10). The LC2029 and LS7247 consortium did not affect IL-8 and TNF-α production compared with the intact control. The preliminary introduction of the consortium into the enterocyte culture medium canceled CJ-induced increases in IL-8 production (Table 9) and TNF-α production (Table 10) in intestinal epithelial cells.

### 2.7. The Effect of the LC2029 and LS7247 Consortium on Intestinal Barrier Function in Caco-2 Enterocyte Monolayers

All MDR CJ strains dramatically reduced the transepithelial electrical resistance levels, indicating the destruction of the intestinal barrier modeled in vitro by monolayers of Caco-2 enterocytes (Table 11). Monolayers of Caco-2 pretreated with the consortium of LC2029 and LS7247 strains became resistant to the destructive action of CJ.

### 2.8. The Effect of the LC2029 and LS7247 Consortium on Intestinal Paracellular Permeability in Caco-2 Enterocyte Monolayers

All MDR CJ strains caused an increase in the intestinal paracellular permeability (IPP) of Caco-2 monolayers (Table 12). These data also indicated the destructive effect of CJ on the intestinal barrier. The presence of the LC2029 and LS7247 consortium in the Caco-2 monolayer culture medium effectively blocked the CJ-induced increase in IPP.

### 2.9. The Effect of the LC2029 and LS7247 Consortium on Zonulin Secretion in Caco-2 Monolayers

All MDR CJ strains caused an increase in zonulin secretion in Caco-2 monolayers (Table 13). These data additionally indicated the destructive effect of CJ on the intestinal barrier. The presence of the LC2029 and LS7247 consortium in the culture medium of the Caco-2 monolayer effectively blocked CJ-induced zonulin secretion.

### 2.10. The Effect of CFS from the LC2029 and LS7247 Consortium on mRNA Expression of Intestinal Alkaline Phosphatase in Caco-2 Enterocytes

All MDR CJ strains significantly decreased the IAP mRNA expression in Caco-2 monolayers (Table 14). CFS from the LC2029 and LS7247 consortium increased the IAP mRNA expression in Caco-2 monolayers. Preliminary addition of CFS to a culture medium with Caco-2 monolayers prevented the CJ-dependent suppression of IAP mRNA expression.

### 2.11. Study of Tolerance of LC2029 and LS7247 Strains Included in the Consortium

The results of the in vitro assessment of LC2029’s tolerance to gastric and intestinal stresses are presented in Table 15. According to the data obtained, LC2029 is highly resistant to gastric stresses. After 60 min of exposure to gastric juice, the degree of resistance (RD) of LC2029 to gastric stress was 1.2 ± 0.1. After 5 h of exposure to intestinal juice, the RD of LC2029 to intestinal stress was 7.7 ± 0.3.

We previously studied the tolerance of LS7247 to gastric and intestinal stresses in vitro, before the inclusion of this strain in the anti-CJ consortium. The results of this research were published in [108]. LS7247 is highly resistant to gastric stress. After 60 min of exposure to gastric juice, the RD of LS7247 to gastric stress was 1.1 ± 0.1. After 5 h of exposure to intestinal juice, the RD of LS7247 to intestinal stress was RD = 4.0 ± 0.3. Thus, the strains LC2029 and LS7247, which are part of the created synbiotic (LC2029, LS7247, and the prebiotic Actigen), are resistant to gastric and intestinal stresses.

### 2.12. Study of the Adhesion of the LC2029 and LS7247 Strains Included in the Consortium to Human and Animal Enterocytes

The adhesion of LC2029 to human Caco-2, porcine IPEC-J2, chicken CPCE, and bovine BPCE enterocytes is shown in Table 16. LC2029 demonstrated efficient adhesion to human immortalized Caco-2 intestinal epithelial cells (adhesion activity (AA): 100%; adhesion index (AI): 49 ± 6), porcine immortalized IPEC-J2 intestinal epithelial cells (AA: 100%; AI: 51 ± 8), chicken primary CPCE intestinal epithelial cells (AA: 100%; AI: 45 ± 4), and bovine primary BPCE intestinal epithelial cells (AA: 100%; AI: 47 ± 5). These findings indicate that LC2029 exhibited high adhesive ability to enterocytes regardless of the host species (human, pig, chicken, or bovine).

We previously studied the adhesion of LS7247 to human and animal enterocytes in vitro, before the inclusion of this strain in the anti-CJ consortium. The results of this research were published in [108]. LS7247 showed high adhesive ability to human and animal enterocytes. Strong attachment of the LC2029 and LS7247 consortium to human and animal enterocytes is necessary to ensure resistance of the intestine against colonization CJ, and to perform various probiotic functions in the digestive system.

## 3. Discussion

Among scientists dealing with the problem of campylobacteriosis in farm animals, there is a misconception that, unlike its behavior in humans, CJ is commensal, and colonization of the intestine by this microorganism, as well as its carriage, proceeds without any clinical manifestations of the disease [100,114,115,116,117]. Nevertheless, information is accumulating about the occurrence of focal liver necrosis and the appearance of signs of disseminated hemorrhagic gastroenteritis in infected chickens [118]. CJ can cause NEC in hatched chicks, and it persists in the blood, muscles, and internal organs (spleen, liver, thymus) of chickens [117,119]. In response to intestinal infection, the chicken’s innate immunity triggers the production of pro-inflammatory cytokines that disrupt the barrier function of the GIT [120]. The aforementioned data indicate that campylobacteriosis is a zooanthroponous, socially significant infection of humans and animals. The creation of large poultry farms around the world and the use of antibiotics as growth factors have allowed CJ to adapt to the intestines of industrial poultry and enter the human intestine via the food chain [103,121]. The optimal growth of CJ is observed at 42 °C. Due to their higher body temperature, poultry are among the most common food animals that are carriers of CJ, representing the main source of infection for humans [122].

CJ adhesion is the first step to attacking human and animal enterocytes and initiating an infectious process. The specific interaction of CJ with host enterocytes occurs with the help of adhesins located in the OM on the surface of the pathogen. Unlike the adhesion mechanisms implemented by *Escherichia coli* and *Salmonella*, in CJ, pili and fimbria do not participate in interactions with enterocytes [123,124,125]. The most studied CJ adhesins are Cad and FlpA, which ensure the interaction of CJ with the enterocyte fibronectin, as well as JlpA (jejuni lipoprotein A), which interacts with the heat shock protein Hsp 90 [126,127,128,129]. CJ uses these adhesins to attach to enterocytes, significantly complicating the elimination of the pathogen from the host organism both through intestinal promotion and its displacement with the help of probiotics. Thus, the specific recognition of host enterocytes and the successful adhesion of CJ to them is a central factor for effective colonization and invasion, as well as the persistence of the pathogen and the development of campylobacteriosis [124,130,131,132]. Targeted inhibition of CJ adhesion by anti-adhesive agents enables prevention or intervention at the earliest stages of disease development, as demonstrated in experiments using uropathogenic *E. coli* [133,134] and *Helicobacter pylori* [135,136]. 

Human milk oligosaccharides (HMOs) are natural prebiotics that form a healthy intestinal microbiota in infants [137,138]. In vivo experiments in a mouse model have shown that HMOs block CJ adhesion and pathogen colonization on host enterocytes [139,140]. HMOs reduced the invasion of CJ into cultured Hep-2 and HT-29 cells and inhibited the production of pro-inflammatory cytokines in vitro [141]. These results were confirmed by data from an epidemiological study in which it was found that HMOs were able to prevent CJ-induced diarrhea in breastfed infants [139]. However, the practical application of HMOs as a prebiotic requires the organization of large-scale production. As an alternative to antibiotics to prevent and reduce CJ colonization in chickens, prebiotics such as mannan oligosaccharides, inulin, and oligofructose have also been studied [142,143,144]. The results are contradictory. The prebiotic Actigen, derived from the mannan-rich fraction of the *Saccharomyces cerevisiae* yeast cell wall, is used as a feed additive for animals at a dosage of 400 gm per ton [145]. Actigen blocks the adhesion of intestinal pathogens (*Salmonella*, *Campylobacter*) to enterocytes [146,147], stimulates intestinal mucosa development and integrity, produces short-chain fatty acids, and positively influences microbiome uniformity, productivity-associated taxa, and laying performance [147,148,149]. The prebiotic Actigen and CFS from the consortium showed an additive effect in inhibiting the adhesion of CJ to human Caco-2 (Table 1), pig IPEC-J2 (Table 2), chicken CPCE (Table 3), and bovine BPCE (Table 4) enterocytes. The prebiotic Actigen and Slp2 secreted by the LC2029 strain also showed an additive effect in inhibiting the adhesion of CJ to human Caco-2 cells (Table 5).

The bactericidal effect of the LC2029 and LS7247 consortium on CJ is realized in two stages. In the first stage, permeabilization of CJ’s OM is carried out (Table 6). LC2029’s genome contains genes responsible for the production of permeabilizers such as Slp2 and the bacteriocin helveticin M [150,151,152]. LS7247’s genome contains genes responsible for the production of lactic acid as a permeabilizer [153].

The OM of Gram-negative pathogens (*E. coli*, *Salmonella*, *Campylobacter*) provides their cells with an effective barrier against the permeation of external harmful agents, including antibiotics. The OM of CJ consists of lipooligosaccharides (LOSs), which contain sialic acid (N-acetylneuraminic acid, N-AcNeu) [154]. LOSs consist of the lipid A and the core oligo- and polysaccharides, which, like the CJ capsule, can be structurally diverse [155]. The poor permeability of the OM creates problems in the treatment of infectious disease caused by this pathogen. The permeabilizers of the OM of Gram-negative pathogens are polycations and chelators [156]. Polycations are capable, under certain conditions, of binding to the anionic centers of LOSs. Such polycations include various cationic leukocyte peptides and probiotic bacteriocins (defensins and helveticin-M). The cationic character is not the only determining factor necessary for perforation activity. Slps produced by probiotic lactobacilli are much more effective permeabilizers [151,152]. Chelators (such as EDTA, nitrilotriacetic acid, and lactic acid) destroy the OM by removing Mg^2+^ and Ca^2+^, and they are effective permeabilizers [156]. Lactic acid by itself and lactobacilli that produce it are used to reduce the prevalence of CJ in poultry flocks in agriculture [157,158]. The introduction of lactic acid into chicken carcasses after slaughter reduces the levels of CJ on the surface of commercial poultry meat [159,160]. Lactic acid increases the permeability of CJ’s OM and reduces the pathogen viability [157].

At the second stage of the bactericidal action of the LC2029 and LS7247 consortium on CJ, PG-degrading enzymes penetrate into the CJ cells through a permeable OM to destroy PG. Co-cultivation of LC2029 and LS7247 with CJ reduced the abundance of living pathogen cells by 6 log (Table 7). After permeabilization of CJ’s outer membrane with lactic acid, Slp2, and helveticin-M (the first stage of action of the LC2029 and LS7247 consortium), nisin and enterolysin A from LS7247 are able to penetrate the pathogen cells and begin the destruction of peptidoglycan (PG). The bacterial cell wall of *E. coli*, *Salmonella*, and *Pseudomonas aeruginosa* consists mainly of a rigid PG exoskeleton [161,162]. PG in CJ is additionally O-acetylated [163]. PG is an important molecule for bacterial survival [164]. It counteracts osmotic pressure, maintains the shape and integrity of cells, and also serves as a protective barrier against physical, chemical, and biological threats [161]. PG is located on the outer side of the cytoplasmic membrane of bacteria [165] and is one of the main targets of antibiotics [166], bacteriophages, and lactobacilli producing endolysins [167]. Nisin has a bactericidal effect against CJ, *E. coli*, and *Salmonella*; however, the outer membranes of these pathogens block the penetration of this bacteriocin into the cell. Treatment of pathogens with Slp or reuterin, which increase the permeability of the outer membrane of pathogens, ensures penetration of nisin into the cell and the manifestation of its bactericidal action [151,152,168]. Currently, nisin is considered to be a promising bacteriocin as a therapeutic agent for several key Gram-negative pathogens of humans and animals, including CJ [169]. The lipid component II of CJ’s bacterial cell wall is a receptor for nisin [170,171]. It forms a complex with a lipid II precursor, which is involved in the construction of PG around the cell and disrupts the formation of the cell wall. Nisin also targets lipid II as a docking station for the formation of pores. This leads to the efflux of cellular constituents [172]. The enterolysin A molecule from LS7247 contains lysozyme and metalloendopeptidase, responsible for the destruction of short peptides connecting linear chains of glycans [108]. Lysozyme catalyzes the hydrolysis of the β-1,4 glycoside bond between N-acetylmuramic acid and N-acetylglucosamine in the PG of the bacterial cell wall [173,174]. The mechanisms of PG biosynthesis in CJ include the O-acetylation process. This modification gives CJ resistance to lysozyme [163,175,176]. To overcome this resistance, an increase in the concentration of lysozyme and an extension of its use period are required. In vivo experiments using weaned suckling pigs showed that lysozyme reduced the levels of campylobacter and other pathogens [177]. Piglets weaned from a sow at the age of 21 days received milk for eight weeks from transgenic goats expressing human lysozyme at 68%, the level found in human milk and the milk of young sows, as feeding subjects. The levels of Firmicutes (Clostridia) declined whereas those of Bacteroidetes increased over time in response to the consumption of lysozyme-rich milk. The numbers of *Bifidobacteriaceae* and *Lactobacillaceae* increased, and the numbers of bacteria associated with infections (*Campylobacterales*, *Mycobacteriaceae*, *Streptococcaceae*) decreased. This study demonstrated the ability of lysozyme to control the composition of the intestinal microbiota, enriching it with beneficial microbes and reducing the number of harmful microbes. Lysozyme stimulates the production of IgA in the intestine [178] and contributes to the preservation of weight and growth of young animals during weaning [179].

The CJ strains used in our work stimulated TLR4 mRNA expression in Caco-2 enterocytes in vitro (Table 8). Increased TLR4 mRNA expression in Caco-2 enterocytes led to the stimulation of pro-inflammatory cytokine (IL-8 and TNF-α) production (Table 9 and Table 10). Our results are consistent with the data of Ayllón et al. (2017), who also noted that human enterocytes react with intense inflammation to the CJ pathogen [180]. Toll-like receptors (TLRs) are necessary to protect the body, but evidence has now accumulated that they are associated with the pathogenesis of inflammatory diseases [181,182,183]. TLR4 is a receptor of innate immunity that stimulates inflammation in response to LOSs produced by bacteria, particularly CJ [184,185]. LOSs are TLR4 agonists [186]. Via NF-κB activation, TLR4 contributes to the activation of the cytosolic inflammasome [187,188,189,190,191]. Dephosphorylation of LOSs leads to the cancellation of their ability to act as a TLR4 agonist and cause pro-inflammatory reactions. The dephosphorylation of LOSs is carried out by the IAP enzyme [192]. The LC2029 and LS7247 consortium controls IAP production in enterocytes (Table 14). In vivo experiments on gnotobiotic mice deficient in the anti-inflammatory cytokine IL-10(−/−) showed that orally injected CJ 43,431 cells colonized the GIT in high concentrations and caused acute enterocolitis, as evidenced by bloody diarrhea and pronounced histopathological changes in the mucous membrane of the colon [193]. Immunopathology was also characterized by an increase in the numbers of T and B lymphocytes and apoptotic enterocytes, as well as increased concentrations of TNF-α and IL-8 in the inflamed colon. Of particular interest is the less pronounced intestinal immunopathology in mice with TLR4(−/−) deficiency [193]. This indicated the ability of LOSs from CJ to provoke excessive activation of the innate immunity of experimental animals. LOSs are a major bacterial factor triggering innate immune responses in human campylobacteriosis [194]. Overly activated innate immunity reduced the colonization resistance of animals without TLR4(−/−) deficiency, which created conditions for the development of campylobacteriosis [194,195,196,197]. Excessive activation of innate immunity is necessary for CJ to destroy the intestinal barrier in humans, increase paracellular permeability, and destroy tight junctions (TJs) between enterocytes [198]. CFS from the LC2029 and LS7247 consortium inhibited the increase in TLR4 mRNA expression induced by CJ in Caco-2 enterocytes (Table 8). CFS inhibited CJ-induced production of the pro-inflammatory cytokines IL-8 and TNF-α in Caco-2 enterocytes (Table 9 and Table 10). We found that CJ strains reduce the intestinal barrier and increase the paracellular permeability and secretion of zonulin in Caco-2 enterocyte monolayers. The LC2029 and LS7247 consortium preserved the intestinal barrier and inhibited the increase in paracellular permeability and zonulin secretion in Caco-2 enterocyte monolayers (Table 11, Table 12 and Table 13). CJ strains inhibited IAP mRNA expression in enterocytes (Table 14). CFS from the LC2029 and LS7247 consortium increased IAP mRNA expression in enterocytes. The IAP enzyme is one of the main molecular factors that ensure intestinal homeostasis [199,200,201]. IAP dephosphorylates LOSs, which leads to the elimination of their properties as TLR4 agonists [192] and increases the efficiency of excretion of enteropathogens from the intestine [202]. The enzyme limits inflammation, prevents the development of a cytokine storm [203], and preserves intestinal barrier functions [204,205]. 

Thus, for the first time, we have created a synergistic synbiotic that inhibits the adhesion of MDR CJ to human and animal enterocytes and has a bactericidal effect on the pathogen. This synbiotic also prevents intestinal barrier dysfunction that disrupts intestinal homeostasis.

## 4. Materials and Methods

### 4.1. Bacterial Strains and Growth Conditions

The bacteria used in this work and the conditions of their reproduction are given in Table 17.

### 4.2. Enterocytes and Growth Conditions

#### 4.2.1. Human Enterocytes

Immortalized Caco-2 cells (model system of the human small intestine in vitro) and immortalized HT-29 cells (model system of the human colon in vitro) were cultured in DMEM medium (HiMedia, Thane, India) containing 10% fetal calf serum and 0.02% penicillin and streptomycin each. The cells were seeded into 12-well cell culture plates at a density of 5 × 10^5^ cells/mL to establish a cell monolayer. The plates were incubated for 48 h at 37 °C under 5% CO_2_. To study the monolayer of Caco-2 cells as an intestinal barrier, each cell line was cultured for 15 days, with daily replacement of medium.

#### 4.2.2. Porcine Enterocytes

Immortalized enterocytes from the porcine intestine, IPEC-J2, were employed as a relevant in vitro model system for porcine intestinal pathogen–host cell interactions. IPEC-J2 cells were cultured following a similar procedure to that described in Section 4.2.1, in DMEM medium.

#### 4.2.3. Chicken Enterocytes

Chicken primary cecal enterocytes (CPCEs) were isolated from the cecum of 2-week-old chickens (Cobb-500 cross) using the protocol presented in [206]. The enterocytes were suspended in DMEM (HiMedia, India) containing 2.5% FCS, 0.1% insulin, 0.5% transferrin, 0.007% hydrocortisone, 0.1% fibronectin, and 0.02% penicillin and streptomycin each, and seeded according to the same procedure described in Section 4.2.1.

#### 4.2.4. Bovine Enterocytes

Bovine primary colon enterocytes (BPCEs) were obtained from the colon of a calf using the method developed in [207]. The BPCEs were grown in DMEM/F12 medium containing 5% fetal bovine serum (FBS), 1% streptomycin–penicillin, 1% L-glutamine, 0.1% epidermal growth factor (EGF), and 0.1% each of insulin and human transferrin. 

### 4.3. Preparation of CFS and ∆CFS from the Consortium of LC2029 and LS7247 Strains

Native CFS was prepared from culture of the LC2029 and LS7247 consortium, as previously described in [208]. Briefly, the LC2029 and LS7247 strains were grown separately overnight in Man–Rogosa–Sharpe (MRS) broth under anaerobic conditions at 37 °C. Overnight culture of each strain was used to obtain a consortium containing 1 × 10^8^ CFU/mL of LC2029 and 1 × 10^8^ CFU/mL of LS7247 in the MRS broth, and then grown anaerobically for 48 h. CFS was obtained using centrifugation at 5000× *g* for 20 min at 4 °C, filter-sterilized using a 0.22 µm pore size filter (Millipore, Billerica, MA, USA), and concentrated using an RVC2-18 Rotational Vacuum Concentrator (Christ, Osterode am Harz, Germany) through speed-vacuum drying. Lyophilized sediment of the CFS from the consortium of LC2029 and LS7247 strains was used in the experiments. ∆CFS was obtained by incubating CFS with proteinase K at 37 °C for 60 min. The concentration of proteinase K was 2 µg/mL.

### 4.4. Slp2 Isolation from the LC2029 Strain

Slp2 was isolated from the LC2029 strain as described in [209].

### 4.5. Adhesion Investigation

The inhibitory effect of the CFS from the consortium of LC2029 and LS7247 strains and the prebiotic Actigen^®^ (Alltech, Inc., Nicholasville, KY, USA) on the adhesion of CJ to human Caco-2, pig IPEC-J2, chicken CPCE, and bovine BPCE enterocytes was determined according to the method described in [208].

### 4.6. Measurement of Extracellular ATP in CJ Strains

Extracellular ATP levels after the treatment of CJ strains with CFS were determined using the ATP detection kit (Beyotime, Nantong, China). Measurement of luminescence was performed using an Infinite 200 PRO microplate reader (Tecan, Männedorf, Switzerland).

### 4.7. Determination of the LC2029 and LS7247 Consortium’s Antibacterial Activity Against CJ

The antagonistic activity of the LC2029 and LS7247 consortium against MDR CJ strains was determined by co-cultivation in TGVC medium (HiMedia, India) at 37 °C, microaerobically (5% O_2_, 10% CO_2_, 85% N_2_), for 48 h (HiMedia, India), as described in [210]. CJ cells (CFU/mL) grown in monoculture (control) and in co-culture with the LC2029 and LS7247 consortium were counted by serial dilutions with seeding on BHI agar (HiMedia, India).

### 4.8. Quantitative Real-Time Polymerase Chain Reaction

Quantitative real-time polymerase chain reaction (qRT-PCR) was carried out as described previously in [208]. The primers used for evaluating the effects of the LC2029 and LS7247 consortium on the expression of TLR4 and IAP genes in Caco-2 enterocytes are shown in Appendix A. The β-actin gene was used as a housekeeping reference gene.

### 4.9. Cytokine Quantification

The production of IL-8 and TNF-α was measured in supernatants of enterocytes that were untreated or treated with CJ (1 × 10^3^ CFU/mL) for 8 h. Production of cytokines was measured with ELISA kits (Thermo Fisher Scientific, Waltham, MA, USA) according to the protocol specified for the specific ELISA kit (Genie, Dublin, Ireland). The sensitivity of all ELISA kits was 2–6 pg/mL.

### 4.10. Transepithelial Electrical Resistance Measurements

Transepithelial electrical resistance was measured using an epithelial volt-ohm-meter (EVOM WPI, Berlin, Germany) to determine cell monolayer integrity. Studies were conducted at each instance of culture medium exchange, according to the manufacturer’s instructions.

### 4.11. Paracellular Permeability Evaluation

The evaluation of paracellular permeability was performed as previously described in [208]. Fluorescein isothiocyanate–dextran with a molecular weight of 4 kDa (FD4) was employed to evaluate the paracellular permeability of the intestinal epithelium layers formed by enterocytes; 250 µL of FD4 solution (1 mg/mL in Hanks’ Balanced Salt Solution (HBSS)) was added to the apical compartment, and 800 µL of HBSS was added to the basolateral compartment. The basolateral side was transferred to a black 96-well plate (Greiner Bio-One, Frickenhausen, Germany) after a 2 h incubation at 37 °C (150 µL). HBSS and FD4 solutions served as negative and positive controls, respectively. A TECAN Infinite M200 plate reader (Tecan Trading AG, Männedorf, Switzerland) was used to measure fluorescence intensity at excitation and emission wavelengths of 490 and 520 nm, respectively.

### 4.12. Quantification of the Tight Junction Regulator Zonulin 

Secretion of the tight junction regulator zonulin in the supernatant by enterocytes was quantified by using specific ELISA kits (AssayGenie, Dublin, Ireland), according to the manufacturer’s instructions.

### 4.13. Statistical Analysis

The results were subjected to analysis using a one-way ANOVA and presented as the means ± standard deviation (SD) from six independent experiments, with each experiment tested in triplicate. Statistical significance was determined using Student’s *t*-test, and findings were considered significant at *p* < 0.05.

## 5. Conclusions

CJ is a highly adapted microorganism that has penetrated the human food chain, an etiological agent of the most common bacterial gastroenteritis, and a serious public health problem. It is necessary to improve the effectiveness of methods for the prevention of campylobacteriosis. Synbiotics, feed additives, and functional nutrition products based on them, which have anti-adhesive activity, open up a new field of application of antibacterial drugs. Our in vitro results showed that the created innovative synbiotic, consisting of a consortium of LC2029 and LS7247 strains and the prebiotic Actigen, is synergistic. Actigen and CFS from the consortium of LC2029 and LS7247 strains showed an additive effect in inhibiting the adhesion of CJ strains to human and animal enterocytes. The consortium had a bactericidal effect on MDR CJ strains. It produced factors that increased the permeability of the outer and inner membranes and factors that degraded PG in MDR CJ strains. In combination with Actigen, the consortium prevented intestinal barrier dysfunction, which disrupts intestinal homeostasis. These data highlight the potential of using a synergistic synbiotic as a preventive strategy for creating feed additives and functional nutrition products based on it to combat the prevalence of campylobacteriosis caused by MDR strains in animals and humans. In the future, we plan to investigate the effectiveness of the created synergistic synbiotic in vivo as a feed additive for the prevention and treatment of campylobacteriosis in experiments on broiler chickens. Special attention will be paid to studying the effect of the created synbiotic on increasing the antagonistic activity of the intestinal microbiota toward MDR strains of CJ.

## Figures and Tables

**Table 1 antibiotics-13-01143-t001:** The effect of the prebiotic Actigen and CFS from the consortium of LC2029 and LS7247 on the adhesion of CJ strains to human Caco-2 enterocytes.

Strains	PBS (Control)	Actigen ^1^	CFS ^2^	MIXT ^3^	∆CFS ^4^	∆MIXT ^5^
*C. jejuni* ATCC 43431	22.7 ± 1.3	7.8 ± 0.6 **	6.2 ± 0.5 **	1.3 ± 0.3 ***	21.5 ± 1.4	8.5 ± 0.7 **
*C. jejuni* IIE PW 7312	39.4 ± 1.1	12.3 ± 0.4 **	10.5 ± 0.6 **	1.6 ± 0.4 ***	38.7 ± 1.3	11.5 ± 0.9 **
*C. jejuni* IIE NB 7318	25.3 ± 1.4	8.4 ± 0.5 **	6.7 ± 0.5 **	1.5 ± 0.3 ***	24.6 ± 1.2	7.7 ± 0.5 **
*C. jejuni* IIE MA 7345	34.5 ± 1.3	10.5 ± 0.6 **	7.4 ± 0.4 **	1.5 ± 0.2 ***	32.4 ± 1.0	9.6 ± 0.6 **
*C. jejuni* IIE MA 7356	49.7 ± 1.6	14.2 ± 0.7 **	11.5 ± 0.6 **	1.7 ± 0.4 ***	48.6 ± 1.1	13.5 ± 0.6 **
*C. jejuni* IIE BR 7358	23.4 ± 1.1	7.6 ± 0.4 **	5.3 ± 0.5 **	1.5 ± 0.3 ***	22.5 ± 1.2	6.4 ± 0.3 **
*C. jejuni* IIE BR 7361	45.6 ± 1.4	13.4 ± 0.6 **	10.2 ± 0.6 **	1.7 ± 0.2 ***	43.9 ± 1.5	12.5 ± 0.5 **
*C. jejuni* IIE PI 7365	26.3 ± 1.5	8.7 ± 0.5 **	6.4 ± 0.3 **	1.6 ± 0.4 ***	25.7 ± 1.3	7.3 ± 0.4 **
*C. jejuni* IIE CO 7384	56.2 ± 1.7	14.5 ± 0.6 **	11.3 ± 0.5 **	1.8 ± 0.5 ***	54.8 ± 1.5	13.2 ± 0.5 **

^1^—Actigen: 40 µg/mL; ^2^—lyophilized CFS: 40 µg/mL; ^3^—mixture of CFS: 20 µg/mL and Actigen: 20 µg/mL); ^4^—lyophilized ∆CFS: 40 µg/mL; ^5^—mixture of Actigen: 20 µg/mL and lyophilized ∆CFS: 20 µg/mL); ** *p* < 0.01 adhesion of CJ pathogen to Caco-2 alone vs. adhesion of CJ pathogen to Caco-2 + Actigen or adhesion of CJ pathogen to Caco-2 + CFS; *** *p* < 0.001 adhesion of pathogen to Caco-2 alone vs. adhesion of CJ pathogen to Caco-2 + Actigen + CFS.

**Table 2 antibiotics-13-01143-t002:** The effects of the prebiotic Actigen and CFS from the consortium of LC2029 and LS7247 on the adhesion of CJ strains to porcine IPEC-J2 enterocytes.

Strains	PBS (Control)	Actigen ^1^	CFS ^2^	MIXT ^3^	∆CFS ^4^	∆MIXT ^5^
*C. jejuni* ATCC 43431	23.5 ± 1.7	8.5 ± 0.7 **	6.4 ± 0.5 **	1.2 ± 0.3 ***	21.9 ± 1.4	7.9 ± 0.6 **
*C. jejuni* IIE PW 7312	32.9 ± 1.4	11.4 ± 0.5 **	8.3 ± 0.4 **	1.6 ± 0.4 ***	30.6 ± 1.7	10.5 ± 0.6 **
*C. jejuni* IIE NB 7318	25.7 ± 1.2	9.2 ± 0.4 **	7.8 ± 0.5 **	1.5 ± 0.3 ***	23.8 ± 1.3	8.6 ± 0.4 **
*C. jejuni* IIE MA 7345	32.8 ± 1.5	10.6 ± 0.5 **	8.5 ± 0.3 **	1.2 ± 0.2 ***	30.9 ± 1.8	9.4 ± 0.5 **
*C. jejuni* IIE MA 7356	44.7 ± 1.5	13.5 ± 0.7 **	9.4 ± 0.8 **	1.8 ± 0.5 ***	41.7 ± 2.5	12.5 ± 0.6 **
*C. jejuni* IIE BR 7358	22.9 ± 1.2	8.4 ± 0.3 **	7.5 ± 0.4 **	1.1 ± 0.2 ***	21.6 ± 1.2	7.2 ± 0.3 **
*C. jejuni* IIE BR 7361	47.5 ± 1.6	15.3 ± 0.9 **	9.7 ± 0.6 **	2.3 ± 0.4 ***	46.2 ± 2.3	14.6 ± 0.5 **
*C. jejuni* IIE PI 7365	27.6 ± 1.2	9.8 ± 0.5 **	7.3 ± 0.2 **	1.3 ± 0.3 ***	25.6 ± 1.8	8.1 ± 0.4 **
*C. jejuni* IIE CO 7384	50.4 ± 1.8	17.6 ± 0.9 **	10.2 ± 0.9 **	2.8 ± 0.6 ***	48.3 ± 1.6	15.4 ± 0.8 **

^1^—Actigen: 40 µg/mL; ^2^—lyophilized CFS: 40 µg/mL; ^3^—mixture of CFS: 20 µg/mL and Actigen: 20 µg/mL); ^4^—lyophilized ∆CFS: 40 µg/mL; ^5^—mixture of Actigen: 20 µg/mL and lyophilized ∆CFS: 20 µg/mL); ** *p* < 0.01 adhesion of CJ pathogen to Caco-2 alone vs. adhesion of CJ pathogen to Caco-2 + Actigen or adhesion of CJ pathogen to Caco-2 + CFS; *** *p* < 0.001 adhesion of pathogen to Caco-2 alone vs. adhesion of CJ pathogen to Caco-2 + Actigen + CFS.

**Table 3 antibiotics-13-01143-t003:** The effects of the prebiotic Actigen and CFS from the consortium of LC2029 and LS7247 on the adhesion of CJ strains to chicken CPCE enterocytes.

Strains	PBS (Control)	Actigen ^1^	CFS ^2^	MIXT ^3^	∆CFS ^4^	∆MIXT ^5^
*C. jejuni* ATCC 43431	25.9 ± 1.2	9.7 ± 0.4 **	7.8. ± 0.6 **	1.4 ± 0.3 ***	24.2 ± 1.8	8.6 ± 0.4 **
*C. jejuni* IIE PW 7312	40.5 ± 1.6	11.4 ± 0.7 **	8.3 ± 0.5 **	1.8 ± 0.4 ***	38.6 ± 1.5	10.2 ± 0.8 **
*C. jejuni* IIE NB 7318	21.7 ± 1.5	8.5 ± 0.5 **	6.2 ± 0.3 **	1.2 ± 0.3 ***	20.4 ± 1.2	7.3 ± 0.6 **
*C. jejuni* IIE MA 7345	38.9 ± 1.3	10.7 ± 0.4 **	7.4 ± 0.5 **	1.3 ± 0.2 ***	36.5 ± 1.4	9.2 ± 0.8 **
*C. jejuni* IIE MA 7356	46.5 ± 1.4	12.2 ± 0.8 **	8.6 ± 0.3 **	1.4 ± 0.3 ***	44.7 ± 1.6	11.5 ± 0.6 **
*C. jejuni* IIE BR 7358	26.3 ± 1.5	7.2 ± 0.3 **	5.4 ± 0.6 **	1.3 ± 0.2 ***	25.4 ± 1.2	6.8. ± 0.3 **
*C. jejuni* IIE BR 7361	52.6 ± 1.5	14.3 ± 0.7 **	8.9 ± 0.5 **	2.1 ± 0.6 ***	50.8 ± 1.7	13.5 ± 0.4 **
*C. jejuni* IIE PI 7365	31.5 ± 1.3	9.5 ± 0.4 **	6.8 ± 0.3 **	1.5 ± 0.4 ***	29.8 ± 1.5	8.4 ± 0.6 **
*C. jejuni* IIE CO 7384	48.5 ± 1.5	12.7 ± 0.6 **	9.1 ± 0.4 **	2.2 ± 0.3 ***	46.4 ± 1.2	11.5 ± 0.5 **

^1^—Actigen: 40 µg/mL; ^2^—lyophilized CFS: 40 µg/mL; ^3^—mixture of CFS: 20 µg/mL and Actigen: 20 µg/mL); ^4^—lyophilized ∆CFS: 40 µg/mL; ^5^—mixture of Actigen: 20 µg/mL and lyophilized ∆CFS: 20 µg/mL); ** *p* < 0.01 adhesion of CJ pathogen to Caco-2 alone vs. adhesion of CJ pathogen to Caco-2 + Actigen or adhesion of CJ pathogen to Caco-2 + CFS; *** *p* < 0.001 adhesion of pathogen to Caco-2 alone vs. adhesion of CJ pathogen to Caco-2 + Actigen + CFS.

**Table 4 antibiotics-13-01143-t004:** The effects of the prebiotic Actigen and CFS from the consortium of LC2029 and LS7247 on the adhesion of CJ strains to bovine BPCE enterocytes.

Strains	PBS (Control)	Actigen ^1^	CFS ^2^	MIXT ^3^	∆CFS ^4^	∆MIXT ^5^
*C. jejuni* ATCC 43431	21.6 ± 1.4	6.5 ± 0.2 **	5.3 ± 0.4 **	1.2 ± 0.3 ***	19.7 ± 1.8	5.9 ± 0.4 **
*C. jejuni* IIE PW 7312	35.3 ± 1.5	9.4 ± 0.5 **	7.2 ± 0.4 **	1.3 ± 0.4 ***	33.9 ± 1.7	8.4 ± 0.3 **
*C. jejuni* IIE NB 7318	19.4 ± 1.3	5.8 ± 0.2 **	4.7 ± 0.3 **	1.1 ± 0.2 ***	18.6 ± 0.9	4.5 ± 0.3 **
*C. jejuni* IIE MA 7345	37.5 ± 1.6	10.4 ± 0.4 **	7.5 ± 0.5 **	1.4 ± 0.3 ***	36.5 ± 1.6	9.4 ± 0.5 **
*C. jejuni* IIE MA 7356	41.8 ± 1.4	12.5 ± 0.6 **	9.3 ± 0.4 **	1.6 ± 0.3 ***	39.7 ± 2.1	11.2 ± 0.5 **
*C. jejuni* IIE BR 7358	28.7 ± 1.3	7.2 ± 0.3 **	5.5 ± 0.3 **	1.2 ± 0.2***	26.5 ± 1.4	6.4 ± 0.3 **
*C. jejuni* IIE BR 7361	49.6 ± 1.8	13.5 ± 0.4 **	8.3 ± 0.4 **	1.8 ± 0.3 ***	47.4 ± 1.6	12.3 ± 0.7 **
*C. jejuni* IIE PI 7365	24.9 ± 1.2	6.3 ± 0.4 **	5.2 ± 0.7 **	1.2 ± 0.3 ***	23.5 ± 1.1	5.8 ± 0.2 **
*C. jejuni* IIE CO 7384	53.7 ± 1.9	14.8 ± 0.3 **	11.5 ± 0.6 **	2.1 ± 0.4 ***	52.6 ± 1.5	13.4 ± 0.7 **

^1^—Actigen: 40 µg/mL; ^2^—lyophilized CFS: 40 µg/mL; ^3^—mixture of CFS: 20 µg/mL and Actigen: 20 µg/mL); ^4^—lyophilized ∆CFS: 40 µg/mL; ^5^—mixture of Actigen: 20 µg/mL and lyophilized ∆CFS: 20 µg/mL); ** *p* < 0.01 adhesion of CJ pathogen to Caco-2 alone vs. adhesion of CJ pathogen to Caco-2 + Actigen or adhesion of CJ pathogen to Caco-2 + CFS; *** *p* < 0.001 adhesion of pathogen to Caco-2 alone vs. adhesion of CJ pathogen to Caco-2 + Actigen + CFS.

**Table 5 antibiotics-13-01143-t005:** The effects of Slp2 from LC2029 and the prebiotic Actigen on the adhesion of CJ strains to Caco-2 enterocytes.

Strains	PBS (Control)	Slp2 ^1^	Actigen ^2^	MIXT ^3^	∆Slp2 ^4^	∆MIXT ^5^
*C. jejuni* ATCC 43431	24.3 ± 1.5	5.3 ± 0.5 **	9.2 ± 0.5 **	1.0 ± 0.2 ***	25.7 ± 1.2	9.7 ± 0.4 **
*C. jejuni* IIE PW 7312	37.8 ± 1.2	8.4 ± 0.3 **	11.7 ± 0.5 **	1.4 ± 0.3 ***	36.4 ± 1.3	12.4 ± 0.8 **
*C. jejuni* IIE NB 7318	26.5 ± 1.3	5.6 ± 0.3 **	8.6 ± 0.3 **	1.3 ± 0.2 ***	27.8 ± 1.2	9.5 ± 0.6 **
*C. jejuni* IIE MA 7345	31.6 ± 1.2	6.2 ± 0.4 **	11.8 ± 0.4 **	1.2 ± 0.3 ***	34.3 ± 1.1	10.4 ± 0.5 **
*C. jejuni* IIE MA 7356	45.2 ± 1.5	9.7 ± 0.5 **	11.6 ± 0.5 **	1.5 ± 0.3 ***	46.8 ± 1.5	12.5 ± 0.6 **
*C. jejuni* IIE BR 7358	24.6 ± 1.2	6.8 ± 0.3 **	10.3 ± 0.5 **	1.6 ± 0.3 ***	23.7 ± 1.3	9.8 ± 0.5 **
*C. jejuni* IIE BR 7361	46.4 ± 1.5	7.4 ± 0.3 **	10.9 ± 0.5 **	1.5 ± 0.2 ***	47.2 ± 1.6	11.4 ± 0.5 **
*C. jejuni* IIE PI 7365	28.2 ± 1.2	5.2 ± 0.1 **	11.6 ± 0.4 **	1.4 ± 0.3 ***	26.7 ± 1.3	10.5 ± 0.5 **
*C. jejuni* IIE CO 7384	49.3 ± 1.6	9.8 ± 0.4 **	12.3 ± 0.5 **	1.6 ± 0.4 ***	51.8 ± 1.5	11.3 ± 0.6 **

^1^—Slp2: 10 µg/mL; ^2^—Actigen: 40 µg/mL; ^3^—mixture of Slp2: 5 µg/mL and Actigen: 20 µg/mL; ^4^—lyophilized ∆Slp2: 10 µg/mL; ^5^—mixture of lyophilized ∆Slp2: 5 µg/mL and Actigen: 20 µg/mL; ** *p* < 0.01 adhesion of CJ to Caco-2 alone vs. adhesion of CJ to Caco-2 + Slp2 or adhesion of CJ to Caco-2 alone vs. adhesion of CJ to Caco-2 + Actigen; *** *p* < 0.001 adhesion of CJ to Caco-2 alone vs. adhesion of CJ to Caco-2 + Slp2 + Actigen.

**Table 6 antibiotics-13-01143-t006:** The effect of CFS from the LC2029 and LS7247 consortium on the extracellular ATP levels in CJ strains.

Strains	Control ^1^	CFS ^2^
*C. jejuni* ATCC 43431	6.8 ± 0.5	74.8 ± 2.3 **
*C. jejuni* IIE PW 7312	5.7 ± 0.8	42.3 ± 1.1 **
*C. jejuni* IIE NB 7318	5.2 ± 0.3	87.3 ± 1.9 **
*C. jejuni* IIE MA 7345	6.4 ± 0.2	68.7 ± 2.4 **
*C. jejuni* IIE MA 7356	6.3 ± 0.7	65.4 ± 1.8 **
*C. jejuni* IIE BR 7358	5.0 ± 0.4	79.7 ± 2.0 **
*C. jejuni* IIE BR 7361	7.4 ± 0.9	78.2 ± 2.1 **
*C. jejuni* IIE PI 7365	6.5 ± 0.8	85.4 ± 3.2 **
*C. jejuni* IIE CO 7384	7.6 ± 0.5	95.6 ± 3.5 **

^1,2^—Concentration of ATP (nm/OD). ^1^—Control: suspension of CJ cells in intact BHI medium. ^2^—CFS: suspension of CJ cells in CFS from LC2029 and LS7247 consortium; ** *p* < 0.01 extracellular ATP levels in CJ strains (control) vs. CFS. Data are presented as the means ± SD of six independent experiments, tested in triplicate.

**Table 7 antibiotics-13-01143-t007:** The effects of the LC2029 and LS7247 consortium on CJ viability.

Strains	0 h		48 h	
	C ^1^	JC ^2^	C ^1^	JC ^2^
*C. jejuni* ATCC 43431	2 × 10^7^	2 × 10^7^	8 × 10^8^	<10^2^
*C. jejuni* IIE PW 7312	2 × 10^7^	3 × 10^7^	9 × 10^8^	<10^2^
*C. jejuni* IIE NB 7318	3 × 10^7^	3 × 10^7^	7 × 10^8^	<10^2^
*C. jejuni* IIE MA 7345	1 × 10^7^	2 × 10^7^	8 × 10^8^	<10^2^
*C. jejuni* IIE MA 7356	4 × 10^7^	4 × 10^7^	9 × 10^8^	<10^2^
*C. jejuni* IIE BR 7358	3 × 10^7^	3 × 10^7^	7 × 10^8^	<10^2^
*C. jejuni* IIE BR 7361	2 × 10^7^	2 × 10^7^	8 × 10^8^	<10^2^
*C. jejuni* IIE PI 7365	2 × 10^7^	3 × 10^7^	6 × 10^8^	<10^2^
*C. jejuni* IIE CO 7384	1 × 10^7^	1 × 10^7^	9 × 10^8^	<10^2^

^1^—Control; number of CJ cells in monoculture (CFU/mL). ^2^—Number of CJ cells in co-culture with the LC2029 and LS7247 consortium (CFU/mL).

**Table 8 antibiotics-13-01143-t008:** The effect of the LC2029 and LS7247 consortium on the CJ-induced TLR4 mRNA expression in Caco-2 enterocytes.

Strains	TLR4 mRNA (Fold Change)			
	Control	CFS	CJ	CFS + CJ
*C. jejuni* ATCC 43431	1.7 ± 0.4	1.5 ± 0.3 *	15.4 ± 0.6 ***	2.1 ± 0.5 *
*C. jejuni* IIE PW 7312	1.9 ± 0.5	1.4 ± 0.6 *	18.2 ± 0.5 ***	2.0 ± 0.4 *
*C. jejuni* IIE NB 7318	1.6 ± 0.4	1.3 ± 0.2 *	19.3 ± 0.7 ***	1.8 ± 0.3 *
*C. jejuni* IIE MA 7345	2.1 ± 0.3	1.9 ± 0.2 *	17.5 ± 0.4 ***	2.5 ± 0.3 *
*C. jejuni* IIE MA 7356	2.4 ± 0.3	0.8 ± 0.1 *	18.6 ± 0.5 ***	2.7 ± 0.2 *
*C. jejuni* IIE BR 7358	2.2 ± 0.4	1.9 ± 0.3 *	19.5 ± 0.3 ***	2.8 ± 0.4 *
*C. jejuni* IIE BR 7361	1.8 ± 0.2	1.6 ± 0.2 *	16.8 ± 0.4 ***	2.1 ± 0.3 *
*C. jejuni* IIE PI 7365	2.3 ± 0.3	1.9 ± 0.2 *	18.7 ± 0.5 ***	2.6 ± 0.2 *
*C. jejuni* IIE CO 7384	1.6 ± 0.2	1.5 ± 0.2 *	20.6 ± 0.8 ***	1.9 ± 0.2 *

* *p* > 0.05—control vs. consortium or control vs. consortium + *C. jejuni* strain; *** *p* < 0.001—*C. jejuni* strain vs. control.

**Table 9 antibiotics-13-01143-t009:** The effect of the LC2029 and LS7247 consortium on the CJ-induced increase in IL-8 production in Caco-2 enterocytes.

Strains	IL-8 (pg/mL)			
	Control	Consortium	CJ	Consortium + CJ
*C. jejuni* ATCC 43431	40.5 ± 1.2	38.4 ± 2.3 *	125.4 ± 2.6 **	45.1 ± 3.5 *
*C. jejuni* IIE PW 7312	36.7 ± 1.4	41.8 ± 1.9 *	132.5 ± 2.7 **	48.0 ± 3.2 *
*C. jejuni* IIE NB 7318	38.2 ± 1.6	36.5 ± 2.2 *	136.3 ± 2.5 **	42.7 ± 3.6 *
*C. jejuni* IIE MA 7345	42.3 ± 1.5	34.7 ± 2.5 *	157.6 ± 2.9 **	47.5 ± 3.3 *
*C. jejuni* IIE MA 7356	45.1 ± 1.8	37.2 ± 2.8 *	126.8 ± 2.4 **	52.7 ± 5.2 *
*C. jejuni* IIE BR 7358	37.5 ± 1.4	41.9 ± 2.7 *	119.5 ± 1.8 **	46.8 ± 3.4 *
*C. jejuni* IIE BR 7361	29.4 ± 1.7	35.6 ± 2.4 *	135.8 ± 2.3 **	36.5 ± 3.1 *
*C. jejuni* IIE PI 7365	43.6 ± 1.3	46.8 ± 2.5 *	138.7 ± 2.5 **	51.4 ± 4.9 *
*C. jejuni* IIE CO 7384	35.8 ± 1.4	42.5 ± 2.8 *	149.6 ± 2.8 **	43.7 ± 4.1 *

* *p* > 0.05—control vs. consortium or control vs. consortium + CJ; ** *p* < 0.01—CJ vs. control.

**Table 10 antibiotics-13-01143-t010:** The effect of the LC2029 and LS7247 consortium on the CJ-induced increase in TNF-α production in Caco-2 enterocytes.

Strains	TNF-α (pg/mL)			
	Control	Consortium	CJ	Consortium + CJ
*C. jejuni* ATCC 43431	21.5 ± 0.9	18.4 ± 2.0 *	108.7 ± 2.2 **	23.8 ± 3.5 *
*C. jejuni* IIE PW 7312	24.7 ± 1.1	23.5 ± 1.4 *	96.3 ± 1.9 **	26.5 ± 3.4 *
*C. jejuni* IIE NB 7318	16.3 ± 1.2	17.8 ± 2.3 *	95.6 ± 2.1 **	20.7 ± 3.9 *
*C. jejuni* IIE MA 7345	23.8 ± 1.3	24.1 ± 2.7 *	104.8 ± 2.0 **	26.4 ± 3.5 *
*C. jejuni* IIE MA 7356	25.6 ± 1.4	22.7 ± 2.3 *	117.4 ± 2.5 **	29.8 ± 4.2 *
*C. jejuni* IIE BR 7358	18.3 ± 1.2	21.6 ± 2.5 *	98.7 ± 2.3 **	24.5 ± 4.1 *
*C. jejuni* IIE BR 7361	15.7 ± 1.3	18.5 ± 2.7 *	106.8 ± 2.1 *	19.6 ± 4.3 *
*C. jejuni* IIE PI 7365	14.5 ± 0.8	16.3 ± 2.1 *	93.5 ± 1.7 **	18.2 ± 4.5 *
*C. jejuni* IIE CO 7384	22.3 ± 1.1	20.4 ± 2.2 *	107.8 ± 1.9 **	25.7 ± 3.8 *

* *p* > 0.05—control vs. consortium or control vs. consortium + CJ; ** *p* < 0.01—CJ vs. control.

**Table 11 antibiotics-13-01143-t011:** The effect of the LC2029 and LS7247 consortium on the CJ-induced destruction of Caco-2 monolayers.

Strains	Transepithelial Electrical Resistance Value (Ω·cm^2^)			
	Control	Consortium	CJ	Consortium + CJ
*C. jejuni* ATCC 43431	292 ± 11	325 ± 12 *	124 ± 6 **	285 ± 10 *
*C. jejuni* IIE PW 7312	323 ± 12	337 ± 11 *	128 ± 7 **	315 ± 11 *
*C. jejuni* IIE NB 7318	285 ± 10	314 ± 10 *	123 ± 9 **	274 ± 12 *
*C. jejuni* IIE MA 7345	316 ± 8	332 ± 9 *	136 ± 7 **	298 ± 13 *
*C. jejuni* IIE MA 7356	298 ± 9	319 ± 10 *	124 ± 8 **	284 ± 12 *
*C. jejuni* IIE BR 7358	312 ± 7	325 ± 9 *	134 ± 6 **	297 ± 11 *
*C. jejuni* IIE BR 7361	307 ± 9	319 ± 10 *	132 ± 5 **	295 ± 10 *
*C. jejuni* IIE PI 7365	318 ± 6	343 ± 9 *	147 ± 8 **	304 ± 12 *
*C. jejuni* IIE CO 7384	298 ± 10	316 ± 8 *	125 ± 9 **	282 ± 13 *

* *p* > 0.05—control vs. LC2029 and LS7247 consortium or control vs. LC2029 and LS7247 consortium + CJ strain; ** *p* < 0.01—CJ strain vs. control.

**Table 12 antibiotics-13-01143-t012:** The effect of the LC2029 and LS7247 consortium on the CJ-induced increase in IPP in Caco-2 monolayers.

Strains	Paracellular Permeability (%)			
	Control	Consortium	CJ	Consortium + CJ
*C. jejuni* ATCC 43431	4.1 ± 0.2	3.5 ± 0.3 *	45.6 ± 3.1 **	5.5 ± 0.2 *
*C. jejuni* IIE PW 7312	4.3 ± 0.2	3.7 ± 0.4 *	42.8 ± 3.5 **	6.1 ± 0.3 *
*C. jejuni* IIE NB 7318	3.5 ± 0.1	3.1 ± 0.2 *	46.3 ± 2.7 **	5.3 ± 0.1 *
*C. jejuni* IIE MA 7345	3.3 ± 0.2	3.2 ± 0.1 *	41.7 ± 3.4 **	5.7 ± 0.2 *
*C. jejuni* IIE MA 7356	3.1 ± 0.3	3.2 ± 0.1 *	45.6 ± 1.9 **	6.3 ± 0.2 *
*C. jejuni* IIE BR 7358	4.5 ± 0.3	3.9 ± 0.2 *	44.5 ± 3.7 **	6.5 ± 0.1 *
*C. jejuni* IIE BR 7361	3.9 ± 0.4	3.5 ± 0.2 *	42.1 ± 4.5 **	5.4 ± 0.3 *
*C. jejuni* IIE PI 7365	3.7 ± 0.1	3.3 ± 0.2 *	41.8 ± 3.7 **	5.8 ± 0.2 *
*C. jejuni* IIE CO 7384	4.2 ± 0.3	3.6 ± 0.1 *	43.6 ± 4.8 **	5.9 ± 0.3 *

* *p* > 0.05—control vs. LC2029 and LS7247 consortium or control vs. LC2029 and LS7247 consortium + CJ strain; ** *p* < 0.01—CJ strain vs. control.

**Table 13 antibiotics-13-01143-t013:** The effect of the LC2029 and LS7247 consortium on the CJ-induced increase in zonulin secretion by Caco-2 monolayers.

Strains	Zonulin (ng/mL)			
	Control	Consortium	CJ	Consortium + CJ
*C. jejuni* ATCC 43431	2.6 ± 0.2	2.1 ± 0.3 *	25.7 ± 3.1 **	4.5 ± 0.8 *
*C. jejuni* IIE PW 7312	2.3 ± 0.5	2.0 ± 0.4 *	22.6 ± 3.5 **	3.1 ± 0.7 *
*C. jejuni* IIE NB 7318	2.5 ± 0.1	2.1 ± 0.2 *	26.7 ± 2.1 **	3.8 ± 0.3 *
*C. jejuni* IIE MA 7345	2.8 ± 0.2	2.2 ± 0.1 *	31.5 ± 3.9 **	4.7 ± 0.6 *
*C. jejuni* IIE MA 7356	2.1 ± 0.3	1.8 ± 0.1 *	25.7 ± 1.8 **	3.3 ± 0.2 *
*C. jejuni* IIE BR 7358	2.5 ± 0.3	2.3 ± 0.2 *	24.6 ± 3.2 **	3.2 ± 0.4 *
*C. jejuni* IIE BR 7361	2.9 ± 0.4	2.5 ± 0.2 *	26.8 ± 4.3 **	4.4 ± 0.5 *
*C. jejuni* IIE PI 7365	2.7 ± 0.1	1.3 ± 0.2 *	23.5 ± 3.7 **	4.2 ± 0.6 *
*C. jejuni* IIE CO 7384	2.9 ± 0.3	2.5 ± 0.1 *	28.2 ± 4.6 **	4.8 ± 0.3 *

* *p* > 0.05—control vs. LC2029 and LS7247 consortium or control vs. LC2029 and LS7247 consortium + CJ strain; ** *p* < 0.01—CJ strain vs. control.

**Table 14 antibiotics-13-01143-t014:** Impact of CFS from the LC2029 and LS7247 consortium on IAP mRNA expression in Caco-2 monolayers.

Strains	IAP mRNA of Control (%)			
	Control	Consortium	CJ	Consortium + CJ
*C. jejuni* ATCC 43431	100	126 ± 4 *	21 ± 2 **	123 ± 5 *
*C. jejuni* IIE PW 7312	100	138 ± 3 *	35 ± 4 **	126 ± 3 *
*C. jejuni* IIE NB 7318	100	127 ± 4 *	27 ± 3 **	124 ± 5 *
*C. jejuni* IIE MA 7345	100	129 ± 5 *	38 ± 4 **	121 ± 3 *
*C. jejuni* IIE MA 7356	100	143 ± 4 *	25 ± 2 **	129 ± 3 *
*C. jejuni* IIE BR 7358	100	139 ± 6 *	29 ± 3 **	128 ± 4 *
*C. jejuni* IIE BR 7361	100	135 ± 3 *	31 ± 3 **	127 ± 5 *
*C. jejuni* IIE PI 7365	100	152 ± 5 *	26 ± 3 **	145 ± 3 *
*C. jejuni* IIE CO 7384	100	137 ± 3 *	39 ± 3 **	132 ± 4 *

* *p* < 0.05—control vs. consortium or control vs. consortium + CJ; ** *p* < 0.01—CJ vs. control.

**Table 15 antibiotics-13-01143-t015:** Tolerance of LC2029 to gastric and intestinal stresses in vitro.

Gastric Stress *						Intestinal Stress *	
10 min		30 min		60 min		5 h	
CFU/mL		CFU/mL		CFU/mL		CFU/mL	
Experiment, ×10^7^ CFU/mL	Control, ×10^7^ CFU/mL	Experiment, ×10^7^ CFU/mL	Control, ×10^7^ CFU/mL	Experiment, ×10^6^ CFU/mL	Control, ×10^7^ CFU/mL	Experiment, ×10^6^ CFU/mL	Control, ×10^7^ CFU/mL
3.6 ± 0.51	3.7± 0.48	3.0± 0.55	3.5± 0.58	7.7± 0.52	3.6± 0.54	4.5± 0.53	3.5± 0.54
RD = 1.1 ± 0.1 Very good		RD = 1.2 ± 0.1 Very good		RD = 4.7 ± 0.2 Very good		RD = 7.7 ± 0.3 Good	

* Data are presented as the means ± SD of six independent experiments, tested in triplicate.

**Table 16 antibiotics-13-01143-t016:** Adhesion of LC2029 to human, porcine, chicken, and bovine enterocytes.

Enterocytes	LC2029 Adhesion Indicators	
	Adhesion Activity (%)	Adhesion Index *
Human Caco-2	100	49 ± 6
Porcine IPEC-j2	100	51 ± 8
Chicken CPCE	100	45 ± 4
Bovine BPCE	100	47 ± 5

* Data are presented as the means ± SD of six independent experiments, tested in triplicate.

**Table 17 antibiotics-13-01143-t017:** Bacteria used in this study, and their growth conditions.

Bacteria	Strain	Growth Conditions
*L. crispatus*	IIE ^1^ 2029	MRS ^a^ 37 °C anaerobically 24 h
*L. salivarius*	IIE 7247	The same
*C. jejuni*	ATCC ^2^ 43431	BHI ^b^ 37 °C microaerobically (5% O_2_, 10% CO_2_, 85% N_2_) 48 h
*C. jejuni*	IIE PW ^3^ 7312 ^4^	The same
*C. jejuni*	IIE NB ^5^ 7318 ^6^	The same
*C. jejuni*	IIE MA ^7^ 7345 ^8^	The same
*C. jejuni*	IIE MA 7356 ^9^	The same
*C. jejuni*	IIE BR ^10^ 7358 ^11^	The same
*C. jejuni*	IIE BR 7361 ^12^	The same
*C. jejuni*	IIE PI ^13^ 7365 ^14^	The same
*C. jejuni*	IIE CO ^15^ 7384 ^16^	The same

^1^—Collection of microorganisms at the Institute of Immunological Engineering (IIE), Lyubuchany, Moscow Region, Russia. ^2^—American Type Culture Collection, Manassas, VA, USA. ^3^—Pregnant woman. ^4^—Feces of pregnant woman. ^5^—Neonatal baby. ^6^—Feces of neonatal baby with necrotizing enterocolitis. ^7^—Man. ^8^—Feces of man. ^9^—Blood of man. ^10^—Broiler. ^11^—Feces of broiler. ^12^—Feces of broiler. ^13^—Pig. ^14^—Feces of pig. ^15^—Cow. ^16^—Feces of cow. ^a^—MRS (HiMedia, India). ^b^—Brain–heart infusion (BHI) broth supplemented with 0.5% yeast extract or agar-containing BHI plates.

## Data Availability

The data are contained within the article and Appendix A.

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
