# Peer review of "Consortium of Lactobacillus crispatus 2029 and Ligilactobacillus salivarius 7247 Strains Shows In Vitro Bactericidal Effect on Campylobacter jejuni and, in Combination with Prebiotic, Protects Against Intestinal Barrier Dysfunction"

_antibiotics, 2024, doi:10.3390/antibiotics13121143_

Round 1

Reviewer 1 Report

Comments and Suggestions for Authors

Comments on the manuscript „ Consortium of Lactobacillus crispatus 2029 and …”

The authors present a lot of work and describe in depth the in vitro influence of a yeast mannan rich fraction on the interaction of two probiotic bacteria with different cell lines and Campylobacter jejuni. The introduction is a review with about 100 references. It gives a nice introduction into the field. Also the discussion includes a lot of references to interpret the results in the connection to knowledge in the field.

The influence of the culture supernatant of the lactobacteria with and without Actigen on the adherence of 9 C. jejuni strains to enterocytes from human, pig, chicken and cow was determined. The results showed that Actigen enhanced the adherence inhibiting effect of the culture supernantant. 

General Comments

The heading of the article promises too much – it is not shown that it

„... helps maintaining the intestinal homeostasis“ rephrase.

Line 47: similarly: “… its role in protecting intestinal homeostasis” this  was not studied, in vitro studies with certain cell lines were performed which may shed light on the possible in vivo interactions. 

A major problem is that the manuscript contains a huge amount of data on Actigen, a consortium of LC2029 and LS7247 strains and many different Campylobacter jejuni strains, but the source and composition of Actigen is not described. I could not find the reference 113 in PebMed; instead I found a relevant paper which is not in the long reference list: “Yeast mannan rich fraction positively influences microbiome uniformity, productivity associated taxa, and lay performance” by Robert J Leigh , Aoife Corrigan, Richard A Murphy, Jules Taylor-Pickard, Colm A Moran, Fiona Walsh. However, I was disappointed to realize that also in this publication the source and amount of the added mannan rich fraction (MRF) was not given although a detailed analysis of the normal food ingredients was given in the supplement. I think the source and amount of MRF must be given in a scientific publication.

The first results paragraph (lines 163 – 195) is difficult to read – to much numbers and abbreviations. Is there not a different way to present the results and leave the numbers with their standard deviation in the tables? In addition, an experimental outline should be given. It is difficult to find out how the experiments were performed even when one consults the Material and Methods section because you are forced to look up earlier publications. Under paragraph 4.3 mention at least the growth medium and the length of cultivation. What is the difference between CFS and delta-CFS? Also in paragraph 4.5 give a more detailed description.

Line 151: Nisin is not a PG degrading enzyme – rephrase, enterolysin A is a protein with enzymatic activity and not a bacteriocin

Line 177/78: Not CFS but a component in CFS seems to be a protein or a peptide (most likely SLp2 as shown in the next paragraph) – rephrase.

Line 264 explain TEER – and is it necessary to introduce an abbreviation which is only used twice? I think no!

Line 465: “TLR4 is an innate immune factor that causes inflammation in response to LOS produced by bacteria, in particular CJ [179,180]. LOS are TLR4agonists” rephrase, it is a receptor that stimulates…

Line 508: Nisin is not a cell wall degrading factor -   rephrase

In the Materials and Methods section the paragraphs 4.7 and 4.8 are identical – delete one

Reviewer 2 Report

Comments and Suggestions for Authors

The authors of the manuscript Consortium of Lactobacillus crispatus 2029 and Ligilactobacillus salivarius 7247 Strains in Combination with Prebiotic Shows in vitro Anti-Campylobacter Effects and Helps Maintaining the Intestinal Homeostasis developed a symbiotic combining strains of Lactobacillus crispatus 2029 44 (LC2029), Ligilactobacillus salivarius 7247 (LS7247) and the mannan-rich prebiotic Actigen. The authors demonstrated the protective effect of the developed symbiotic in preventing the adhesion of several isolates of Campylobacter jejuni on enterocyte cells of human, pig, bovine and chicken origin. They also demonstrated the inhibition of gene expression of some proinflammatory factors.

Despite the interesting results, the manuscript should be improved:

The studies focused on Campylobacter jejuni, so perhaps such a species should be present in the title.

Abstract:

I suggest rewording it so that it only concerns the results contained in the manuscript and the conclusions resulting from them.

Line 65 and 384 - The results contained in the manuscript only include studies on the inhibition of adhesion to enterocytes. There are no results of studies on the effects on C. jejuni bacteria adsorbed on cells. I cannot agree with the conclusion that the proposed symbiotic will prevent C. jejuni from being virulently in animals or humans. There are also no studies on the effects of long-term symbiotic administration on the microbiome and animals.

The introduction is too extensive for me, I understand that the topic is extensive and touches on many aspects. I proposed shortening it and eliminating repetitions with discussion, e.g. gene expression and protein activity, in my opinion it is better suited for discussion.

Results:

It seems logical to examine the effect of the symbiotic on the bacteria themselves and then demonstrate the effect on adhesion on eukaryotic cells, which is why I would start with chapter 2.4, line 228.

Inhibition of C. jejuni adhesion by the symbiotic includes four cell lines. The remaining studies were performed only on the human Caco-2 line. I would like to ask for justification for choosing only this cell line for further studies.

I would suggest changing the names of the chapters, e.g. it is

Line 163 2.1. The Additive Effect of Prebiotic Actigen and CFS from LC2029 and LS7247 Consortium in Inhibiting CJ Adhesion to Enterocytes .

It should be

The Effect of Prebiotic Actigen and CFS from LC2029 and LS7247 Consortium in CJ Adhesion to Enterocytes

Tables 1-13 Please change the titles of the figures. The titles of the figures contain conclusions that should be in the text as conclusions of the obtained results.

Tables 1S and 2S should be in the full manuscript, especially since they are discussed and cited in the discussion.

 Materials and Methods:

 Lines 524-535 suggest combining subsections into one whole

Lines 553 and 559 repeat the chapter

Conclusions

Suggest rewording to include the most important results obtained from the studies and the villages included in the manuscript.

Round 2

Reviewer 2 Report

Comments and Suggestions for Authors

Dear Authors

Thank you for your explanations and corrections.

I consider this answer:

"Similar data were obtained and general patterns were revealed when studying the effectiveness of the created synbiotic in inhibiting the adhesion of CJ strains to human Caco-2, pig IPEC-J2, chicken CPCE, and bovine BPCE enterocytes. In this regard, in subsequent experiments we used only Caco-2 enterocytes. This cell line is most adapted to study the effect of intestinal bacterial pathogens on innate immunity and the intestinal barrier in vitro."

very satisfactory. And I think that such a mention should be included in the manuscript. In my opinion, this is an important insight that can be used by other researchers in similar studies.

I do not want to enter into a long polemic about the nature of the studied symbiotic. But please take into account that the studies cited by the Authors concern newborns and infants, and the function of glycans in mother's milk, which acts on the digestive tract with a minimal microbiome. It must be taken into account that studies on adults may have a completely different effect, mainly related to the presence of a rich microbiome, which has a task such as the one cited by the Authors, the glycemic effect of mother's milk. And I believe that the Authors' sentence contained in the manuscript:

"This article reflects the first (initial) in vitro stage of studying the properties of a new synbiotic. At the second stage, the preventive efficacy of synbiotic in vivo will be studied in experiments on animals that received synbiotic as a feed additive following infection with C. jejuni, and the effect of synbiotic on the intestinal microbiota will also be studied. Additional data will be obtained on the ability of the new symbiotic to maintain intestinal homeostasis."

This is a very good move